# ProvNeRF: Modeling per Point Provenance in NeRFs as a Stochastic Field

**Kiyohiro Nakayama**[1]    **Mikaela Angelina Uy**[1,2]    **Yang You**[1]    **Ke Li**[3]    **Leonida J. Guibas**[1]

[1] Stanford University    [2] Nvidia    [3] Simon Fraser University

w4756677@stanford.edu    keli@sfu.ca

{mikacuy, yangyou, guibas}@cs.stanford.edu

## Abstract

Neural radiance fields (NeRFs) have gained popularity with multiple works showing promising results across various applications. However, to the best of our knowledge, existing works do not explicitly model the distribution of training camera poses, or consequently the triangulation quality, a key factor affecting reconstruction quality dating back to classical vision literature. We close this gap with ProvNeRF, an approach that models the **provenance** for each point – i.e., the locations where it is likely visible – of NeRFs as a stochastic field. We achieve this by extending implicit maximum likelihood estimation (IMLE) to functional space with an optimizable objective. We show that modeling per-point provenance during the NeRF optimization enriches the model with information on triangulation leading to improvements in novel view synthesis and uncertainty estimation under the challenging sparse, unconstrained view setting against competitive baselines[1].

## 1   Introduction

Neural radiance fields (NeRFs) [42], allowing for learning 3D scenes given only 2D images, have grown in popularity in recent years. It has shown promise in many different applications such as novel view synthesis [4, 6], depth estimation [14], robotics [22, 1], localization [36, 38], etc. Existing literature [11, 15, 45] show that the quality of NeRF reconstruction is correlated with the selection of training camera poses. Similar correlations are observed in the classical literature too, triangulation is highly dependent on camera poses [47, 44, 3], which greatly influences the reconstruction quality. One common and important setting in computer vision literature [2, 37, 40, 16] is the **sparse view** [18] setting in **unconstrained** [53] environments, and triangulation is even more critical, affecting the reconstruction quality as limited input views make the system more sensitive to noise.

Despite the correlation between triangulation and reconstruction quality, to the best of our knowledge, existing works do not explicitly model the former when optimizing the latter. In this work, we address this gap in the literature by modeling for each point the *locations where it is likely visible*. We dub this as the *provenances* of a point. Modeling and learning per-point provenance can help NeRF understand how the training cameras are distributed in space, which inherently links it to triangulation and reconstruction quality.

However, determining the provenances of a point $x$ without the underlying geometry is not straightforward as many factors influence the visibility of each point in the reconstructed geometry. For example, the literature on stereo matching [44, 47] has extensively studied the influences of camera locations on 3D reconstruction. One such well-known challenge arises when selecting the baseline of a pair of cameras in a stereo system. As shown in Fig. 2, points' visibility can suffer from different sets of errors when the length of the camera pair's baseline changes. For NeRFs, the dependence

---

[1]Code will be available at https://github.com/georgeNakayama/ProvNeRF.

38th Conference on Neural Information Processing Systems (NeurIPS 2024).

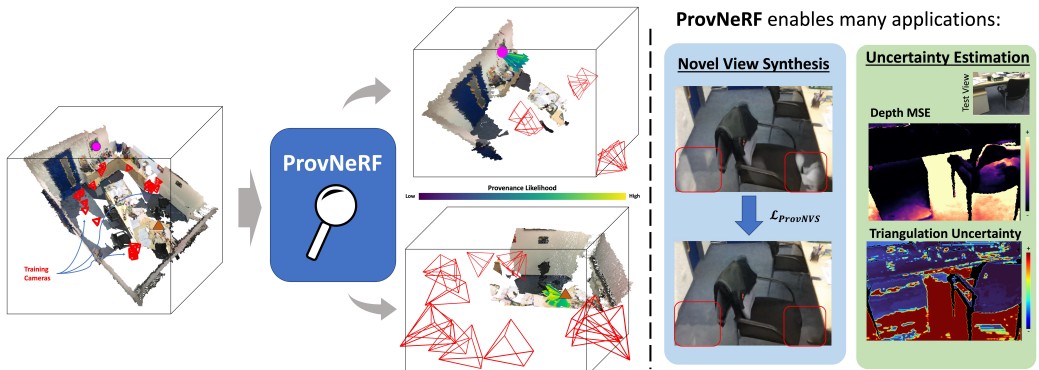

Figure 1: (**Left**) **ProvNeRF** models a provenance field that outputs *provenances* for each 3D point as likely samples (arrows). For 3D points (brown triangle and blue circle), the corresponding provenances (illustrated by the arrows), are locations that likely observe them. (**Right**) **ProvNeRF** enables better novel view synthesis and estimating the uncertainty of the capturing process because it models the locations of likely observations that is critical for NeRF's optimization.

becomes more complex as multiple cameras' visibility needs to be estimated. To overcome this challenge, we propose to model the provenance as the samples from a *probability distribution*, where a location $y$ is assigned with a large likelihood if and only if $x$ is likely to be visible from $y$.

To handle the potential complexity of this distribution, we represent the provenance of $x$ as a set of location *samples*, generated from a learned probability distribution. This is distinct from the existing "attribute" prediction extensions of NeRFs [69, 30, 8] since provenance is a *distribution* for every 3D point in space. Thus, this amounts to modeling an infinite collection of distributions (per-point's provenance) over all 3D points, which is mathematically, a *stochastic field* over $\mathbb{R}^3$. In our work, we extend implicit maximum likelihood estimation (IMLE) [33], a sample-based generative model, to model stochastic fields by adapting the objective to functional space. Furthermore, we derive an equivalent pointwise objective that can be efficiently optimized with gradient descent and use it to model the provenance field.

We dub our method **ProvNeRF** which models per-point provenance during the training stage of NeRF (Fig. 1). This enriches the model with information on triangulation quality when the model parameters are optimized. Once the provenance stochastic field is trained, we show that we can use it to improve novel view synthesis (Sec. 5.1) and estimate triangulation uncertainty in the capturing process (Sec. 5.2) under the challenging sparse, unconstrained view setting.

## 2 Related Works

**NeRFs and their Extensions.** Neural radiance fields (NeRFs) [42] have revolutionized the field of 3D reconstruction [20] and novel view synthesis [50, 32] with its powerful representation of a scene using weights of an MLP that is rendered by volume rendering [41, 58]. Follow-ups on NeRF further tackle novel view synthesis under more difficult scenarios such as unconstrained photo collections [39], unbounded [5], dynamic [35] and deformable [46] scenes, and reflective objects [59, 7]. Going beyond novel view synthesis, the NeRF representation has also shown great promise in different applications such as autonomous driving [56, 62], robotics [1, 22] and editing [67, 63]. Recent works have also extended NeRFs to model other fields in addition to color and opacity such as semantics [69, 68], normals [66], CLIP embeddings [28], image features [30] and scene flow [34]. Most of these works learn an additional function that predicts an auxiliary *deterministic* output at each point that is either a scalar or a vector, trained with extra supervision using volume rendering. All of the above works use a deterministic field to output the additional information. However, because each point's provenance is a probabilistic distribution, we need to model a stochastic field instead of a deterministic field for provenance.

**Sparse View Novel View Synthesis.** NeRFs with rendering supervision alone struggle with sparse view input due to insufficient constraints in volume rendering. Several approaches have been proposed to train NeRFs under the sparse-view regime with regularization losses [43, 64], semantic

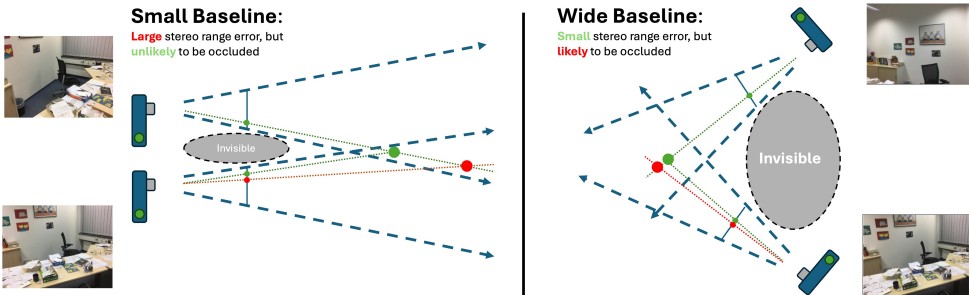

Figure 2: **Complex influence of camera baseline distance on the 3D reconstruction.** *Right:* With a wide baseline, the reconstruction is more robust against 2D measurement noises. However, it is more likely to omit hidden surfaces because the invisible region is larger than a small baseline camera pair. *Left:* With a small baseline, the 3D reconstruction is less likely to suffer from occlusions as the invisible region between cameras is small. However, the reconstruction can be noisy due to large stereo range errors (large deviation in depth with a small amount of noise in the 2D measurement).

consistency [23], and image [60] or cost volume [12, 61, 10] feature constraints. Other works also constrain the optimization using priors from data [24, 65] or depth [49, 57, 54]. Despite addressing the setting with limited number of input views, many works are not specifically designed to tackle our desired sparse, unconstrained views setting as they either focus on object-level [24, 65, 43, 23, 21], limited camera baselines [12, 61], or forward-facing scenes [60] scenes. Recent works [49, 57, 54] have looked into improving the NeRF quality on a more difficult setting of sparse, unconstrained (outward-facing) input views by incorporating depth priors. However, none of these works consider the locations and orientations of training cameras in their optimization process despite it being one of the major factors influencing the NeRF's optimization in a sparse setup.

**Uncertainty Modeling in Neural Radiance Fields.** The current literature separates NeRF's uncertainty into aleatoric – e.g., transient objects or changes in lighting – and epistemic – data limitation due to weak texture or limited camera views – uncertainties. Some works [39, 26] model aleatoric uncertainty by directly predicting the uncertainty values through a neural network. However, their approach requires training on large-scale data and is not suited for estimating the uncertainty of a specific scene. On the other hand, several works explore epistemic uncertainty estimation in NeRFs through variational inference [51, 52, 48], ensemble learning [55, 31], and Bayesian inference [19, 25, 45]. While these works estimate epistemic uncertainty, they still entangle different sources of uncertainty such as texture, camera poses, and model bias, resulting in unclear and inconsistent definitions of the uncertainty quantified. In our work, we specifically model the uncertainty caused by the capturing process that is useful in various downstream tasks [45, 26, 48, 31].

## 3 Preliminaries

### 3.1 Neural Radiance Fields (NeRF)

A neural radiance field (NeRF) is a coordinate-based neural network that learns a field in 3D space, where each point $x \in \mathbb{R}^3$ is of certain *opacity* and *color*. Mathematically, a NeRF is parameterized by two functions representing the two fields $F_{\phi,\psi} = (\sigma_\psi(x), c_\phi(x, d))$, one for opacity $\sigma_\psi : \mathbb{R}^3 \to \mathbb{R}_+$ and one for color $c_\phi : \mathbb{R}^3 \times \mathbb{S}^2 \to [0, 1]^3$, where $d \in \mathbb{S}^2$ is the direction from where $x$ is viewed from. One of the key underpinnings of NeRFs is volume rendering allowing for end-to-end differentiable learning with only training images. Concretely, given a set of $M$ images $I_1, I_2, ..., I_M$ and their corresponding camera poses $P_1, P_2, ..., P_M$, the rendered color of a pixel $x$ is the expected color along a camera ray $r_{i,x}(t) = o_i + t d_{i,x}$, where $o_i$ is the camera origin and $d_{i,x}$ is the ray direction for pixel $x$ that can be computed from the corresponding camera pose $P_i$. The pixel value for 2D coordinate $x$ is then given by the line integral:

$$C_{\phi,\psi}(r_{i,x}) = \int_{t_n}^{t_f} \sigma_\psi(r_{i,x}(t)) T(r_{i,x}(t)) c_\phi(r_{i,x}(t)) \, dt, \tag{1}$$

where $t_n, t_f$ defines the near and far plane, and

$$T\left(\boldsymbol{r}_{i,x}(t)\right) = \exp\left[-\int_{t_n}^{t} \sigma\left(\boldsymbol{r}_{i,x}(s)\right) \, ds\right] \tag{2}$$

is the transmittance of the point $\boldsymbol{r}_{i,x}(t)$ along the direction $\boldsymbol{d}_{i,x}$.

## 3.2 Implicit Maximum Likelihood Estimation

One choice of probabilistic model is implicit maximum likelihood estimation (IMLE) [33] that represents a distribution as a set of samples and is designed to handle possibly multimodal distributions. As an implicit probabilistic model, IMLE learns a parameterized transformation $\boldsymbol{H}_\theta(\cdot)$ of a latent random variable, e.g. a Gaussian $\boldsymbol{z} \sim \mathcal{N}(0, \mathbf{I})$, where $\boldsymbol{H}_\theta(\cdot)$ often takes the form of a neural network that output samples $\boldsymbol{w}_j = \boldsymbol{H}_\theta(\boldsymbol{z}_j)$ with $\boldsymbol{w}_j \sim \mathbb{P}_\theta(\boldsymbol{w})$. Here, $\mathbb{P}_\theta$ is a probability measure obtained by transforming the standard Gaussian distribution measure via $\boldsymbol{H}_\theta$. Given a set of data samples $\{\hat{\boldsymbol{w}}_1, ..., \hat{\boldsymbol{w}}_N\}$, the IMLE objective optimizes the model parameters $\theta$ with

$$\hat{\theta} = \arg\min_\theta \mathbb{E}_{\boldsymbol{z}_1, ..., \boldsymbol{z}_K}\left[\sum_{i=1}^{N} \min_j \|\boldsymbol{H}_\theta(\boldsymbol{z}_j) - \hat{\boldsymbol{w}}_i\|_2^2\right]. \tag{3}$$

It is shown that the above objective to be equivalent to maximizing the likelihood.

## 4 Method

In the following sections, we formally define the provenance at all points as a stochastic field (Sec. 4.1), extend IMLE to model the provenance field (Sec. 4.2), and derive an equivalent pointwise loss for gradient descent (Sec. 4.3).

**Notations.** We denote a stochastic field with a calligraphic font ($\mathcal{D}_\theta$) and samples from the stochastic field using the same letter but bolded ($\boldsymbol{D}_\theta$). Concretely $\boldsymbol{D}_\theta$ is a function sample, a function defined over all points $\boldsymbol{x} \in \mathbb{R}^3$, that is sampled from the stochastic field $\mathcal{D}_\theta$, i.e. $\boldsymbol{D}_\theta \sim \mathcal{D}_\theta$. $\boldsymbol{D}_\theta$ maps each point to one possible sample in its provenances. We also denote the distribution of the provenances at point $\boldsymbol{x}$ as $\mathcal{D}_\theta(\boldsymbol{x})$. Moreover, a provenance sample $\boldsymbol{D}_\theta(\boldsymbol{x})$ from $\mathcal{D}_\theta(\boldsymbol{x})$ is equivalent to evaluating the function $\boldsymbol{D}_\theta \sim \mathcal{D}_\theta$ at $\boldsymbol{x}$. Finally, we let a hat $(\hat{\cdot})$ denote the empirical samples/distributions.

### 4.1 Provenance as a Stochastic Field

The **provenance** of a point is defined as the *locations where it is likely visible from*, and as a point can be visible from multiple locations, it can be represented as samples from a distribution. Specifically, provenances of a point $\boldsymbol{x}$ can be represented as *samples* from its provenance *distribution* $\mathcal{D}_\theta(x)$. That means that the likelihood of sampling a location $\boldsymbol{y} \in \mathbb{R}^3$ from $\mathcal{D}_\theta(x)$ determines how likely $\boldsymbol{x}$ is visible from location $\boldsymbol{y}$. Because such distributions are defined for each 3D point $\boldsymbol{x} \in \mathbb{R}^3$, the collection of per-point provenances forms a stochastic field $\mathcal{D}_\theta$ indexed by coordinates $\boldsymbol{x} \in \mathbb{R}^3$.

Empirically, given sparse training camera views $P_1, \ldots, P_M$, if $\boldsymbol{x}$ is inside the camera frustum $\Pi_i$ for view $P_i$ and is not occluded, an *empirical* sample from the provenances of $\boldsymbol{x}$ can be parameterized as a distance-direction tuple $\hat{D}_i(\boldsymbol{x}) = (\hat{t}_{i,\boldsymbol{x}}, \hat{\boldsymbol{d}}_{i,\boldsymbol{x}}) \in \mathbb{R}_+ \times \mathbb{D}^3$ [2]. Considering all $M$ training views, the empirical distribution of provenances at point $\boldsymbol{x}$ is defined by the following density function:

$$p_{emp}(t, \boldsymbol{d}) = \frac{1}{M}\sum_{i=1}^{M} \delta[(t, \boldsymbol{d}) = (t_{i,\boldsymbol{x}}, \boldsymbol{d}_{i,\boldsymbol{x}})] \tag{4}$$

$$\text{where } (t_{i,\boldsymbol{x}}, \boldsymbol{d}_{i,\boldsymbol{x}}) = \left(v_{i,\boldsymbol{x}} \|\boldsymbol{x} - \boldsymbol{o}_i\|, v_{i,\boldsymbol{x}} \frac{\boldsymbol{x} - \boldsymbol{o}_i}{\|\boldsymbol{x} - \boldsymbol{o}_i\|}\right) \tag{5}$$

where $\delta$ is the Dirac delta function; $v_{i,\boldsymbol{x}} \in [0, 1]$ determines the length of $\boldsymbol{d}$ to handle occlusions. We modeled it as the transmittance from the NeRF model. To recover the location that observes $\boldsymbol{x}$, we can write $\boldsymbol{y}_{i,\boldsymbol{x}} = \boldsymbol{x} - t_{i,\boldsymbol{x}}\boldsymbol{d}_{i,\boldsymbol{x}}$.

---

[2] $\mathbb{D}^3$ denotes a solid ball in $\mathbb{R}^3$

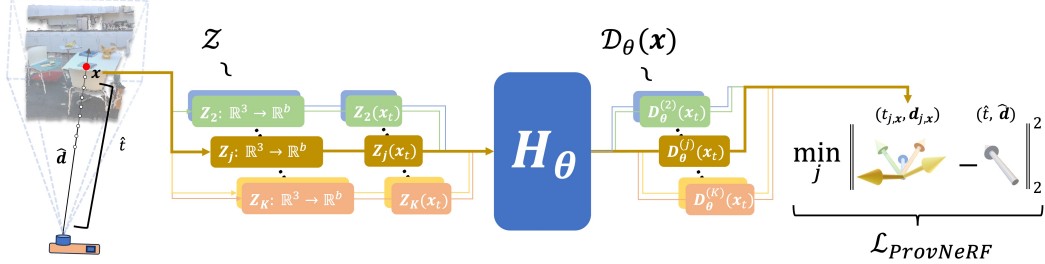

Figure 3: **Training pipeline for ProvNeRF.** For each point $\boldsymbol{x}$ seen from provenance tuple $(\hat{t}, \hat{\boldsymbol{d}})$, with direction $\boldsymbol{d}$ at distance $t$, we first sample $K$ latent random functions $\{\boldsymbol{Z}_j\}$ from distribution $\mathcal{Z}$. The learned transformation $\boldsymbol{H}_\theta$ transforms each $\boldsymbol{Z}_j(\boldsymbol{x})$ to a provenance sample $\boldsymbol{D}_\theta^{(j)}(\boldsymbol{x})$. Finally $\boldsymbol{H}_\theta$ is trained with $\mathcal{L}_{\textbf{ProvNeRF}}$ as defined in Eq. 9.

While the above empirical distribution of provenances is given by the training cameras, the actual distribution of provenances, i.e. for each point the locations that point is likely visible from, can have a more complex dependence on both the underlying geometry and the cameras. To capture this complexity, we model $\mathcal{D}_\theta(\boldsymbol{x})$ as a learnable network, a probabilistic model that can model potentially complex distribution, and one choice of such a model is implicit maximum likelihood estimate (IMLE) [33]. Similar to the empirical distribution, we also represent provenance samples from $\mathcal{D}_\theta(\boldsymbol{x})$ as a distance-direction tuple $(t, \boldsymbol{d})$ as defined in Eq 5. We optimized our network with the empirical distribution $\hat{\mathcal{D}}$ as training signals.

$\mathcal{D}_\theta(\boldsymbol{x})$ defines a distribution for all point $\boldsymbol{x} \in \mathbb{R}^3$. Treating $\mathbb{R}^3$ as the index set, $\mathcal{D}_\theta = \{\mathcal{D}_\theta(\boldsymbol{x})\}_{\boldsymbol{x} \in \mathbb{R}^3}$ defines a stochastic field on $\mathbb{R}^3$ as a collection of distributions $\mathcal{D}_\theta(\boldsymbol{x})$ for all $\boldsymbol{x} \in \mathbb{R}^3$. Because a stochastic field is composed of infinitely many random variables over $\mathbb{R}^3$, existing methods cannot be applied out of the box as they only model finite-dimensional distributions. In the following sections, we extend IMLE [33] to model this stochastic field.

### 4.2 ProvNeRF

**ProvNeRF** models provenances of a NeRF as a stochastic field by extending IMLE [33] to functional space. IMLE learns a mapping that transforms a latent distribution to the data distribution, where each data sample is either a scalar or a vector (Sec. 3.2). However, in our context, since samples from the stochastic field $\mathcal{D}_\theta$ are *functions* mapping 3D locations to provenances, we need to extend IMLE to learn a neural network mapping $\boldsymbol{H}_\theta$ that transforms a pre-defined *latent stochastic field* $\mathcal{Z}$ to the provenance distribution $\mathcal{D}_\theta$ (See Fig. 3).

Let $\mathcal{Z}$ be the stochastic field where each sample $\boldsymbol{Z} \sim \mathcal{Z}$ is a function $\boldsymbol{Z}: \mathbb{R}^3 \to \mathbb{R}^b$. To transform $\mathcal{Z}$ to $\mathcal{D}_\theta$, fIMLE learns a deterministic mapping $\boldsymbol{H}_\theta$ that maps each latent function $\boldsymbol{Z} \sim \mathcal{Z}$ to a function $\boldsymbol{D}_\theta \sim \mathcal{D}_\theta$ via composition: $\boldsymbol{D}_\theta = \boldsymbol{H}_\theta \circ \boldsymbol{Z}$. $\boldsymbol{H}_\theta$ here is represented as a neural network to handle complex transformations from $\mathcal{Z}$ to $\mathcal{D}_\theta$.

We define a latent function sample $\boldsymbol{Z} \sim \mathcal{Z}$ to be the concatenation of a *random linear transformation* of $\boldsymbol{x}$ and $\boldsymbol{x}$ itself. Mathematically, each latent function $\boldsymbol{Z} \sim \mathcal{Z}$ is a block matrix of size $(b+4) \times 3$:

$$\boldsymbol{Z}(\boldsymbol{x}) = \left[\frac{\boldsymbol{z}}{\boldsymbol{I}}\right]\boldsymbol{x}, \text{ where } \boldsymbol{z} \sim \mathcal{N}\left(\boldsymbol{0}, \lambda^2 \boldsymbol{I}\right), \boldsymbol{x} \in \mathbb{R}^3. \tag{6}$$

Although $\boldsymbol{Z}$ can be designed to have non-linear dependence on the input location $\boldsymbol{x}$, we experimentally show that this simple design choice works well across different downstream applications.

To train $\boldsymbol{H}_\theta$, we maximize the likelihood of the training provenances (Eq. 4) under $\mathcal{D}_\theta$ for each $\boldsymbol{x}$ using the IMLE objective [33] extended to functional space. We term this extension as functional Implicit Maximum Likelihood Estimation (fIMLE). Because a direct extension to fIMLE leads to an intractable objective, we derive an efficient pointwise loss between the training provenances and model predictions equivalent to the fIMLE objective in the following section.

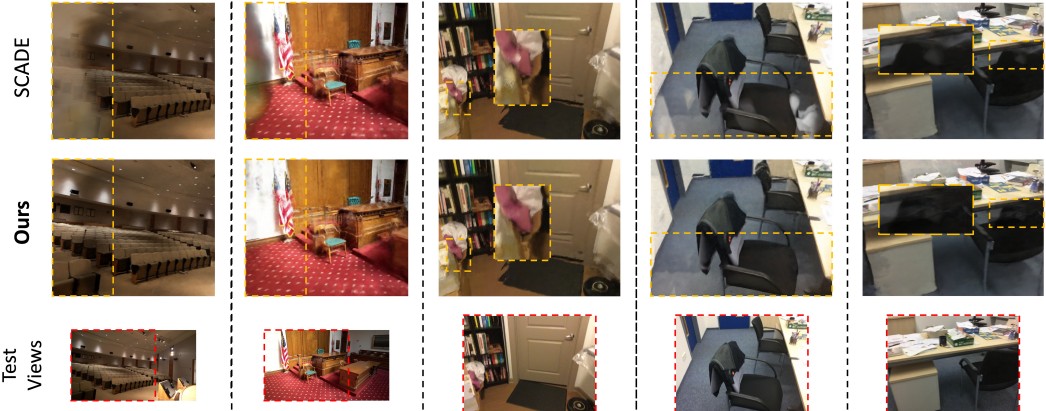

Figure 4: **Visual Effect of $\mathcal{L}_{\textbf{ProvNVS}}$ in Eq. 10**. Compared to pre-trained SCADE model, our method can remove additional floaters in the scene (see the boxed region).

| | **Scannet** | | | **Tanks and Temple** | | |
|---|---|---|---|---|---|---|
| | PSNR ($\uparrow$) | SSIM ($\uparrow$) | LPIPS ($\downarrow$) | PSNR ($\uparrow$) | SSIM ($\uparrow$) | LPIPS ($\downarrow$) |
| NeRF [42] | 19.03 | 0.670 | 0.398 | 17.19 | 0.559 | 0.457 |
| DDP [49] | 19.29 | 0.695 | 0.368 | 19.18 | 0.651 | 0.361 |
| SCADE [57] | 21.54 | 0.732 | 0.292 | 20.13 | 0.662 | 0.358 |
| DäRF [54] | 21.28 | **0.741** | 0.323 | 19.67 | 0.652 | 0.374 |
| **Ours** | **21.73** | 0.733 | **0.291** | **20.36** | **0.663** | **0.349** |

Table 1: **Novel View Synthesis Results.** Our method outperforms baselines in novel view synthesis on both Scannet and Tanks and Temple Datasets. This is because our novel NeRF regularizer in Eq. 10 can remove additional floaters in the scene as shown in Fig. 4. See Sec. 5.1 for details.

### 4.3 Functional Implicit Maximum Likelihood Estimation

We construct an IMLE objective for stochastic fields to maximize the likelihood of training provenances under $\mathcal{D}_\theta$. Similar to Eq. 3, if we have i.i.d. empirical samples $\hat{\boldsymbol{D}}_1, \ldots, \hat{\boldsymbol{D}}_M$ from the empirical stochastic field $\hat{\mathcal{D}}$ (defined in Sec. 4.1), and model samples $\boldsymbol{D}_\theta^{(1)}, \ldots, \boldsymbol{D}_\theta^{(K)}$ from the parameter stochastic field $\mathcal{D}_\theta$, we define the fIMLE objective as

$$\hat{\theta} = \arg\min_\theta \mathbb{E}_{\boldsymbol{D}_\theta^{(1)}, \ldots, \boldsymbol{D}_\theta^{(K)}} \left[ \sum_{i=1}^n \min_j \left\| \hat{\boldsymbol{D}}_i - \boldsymbol{D}_\theta^{(j)} \right\|_{L^2}^2 \right]. \tag{7}$$

Unlike the original IMLE objective (Eq. 3) that can be directly optimized, the fIMLE objective in Eq. 7 requires the computation of a $L^2$ integral norm – a functional analogy to the $L^2$ vector norm – which, in general, is not analytically in closed form. Furthermore, approximations of this integral are very expensive since each point query to $\boldsymbol{D}_\theta$ needs a forward pass through $\boldsymbol{H}_\theta$.

To get around this, we use the calculus of variations to reformulate Eq. 7 to minimize the pointwise difference between the empirical samples and model predictions [3]. This allows us to write the fIMLE objective as

$$\mathcal{L}_{\text{fIMLE}} = \mathbb{E}_{\boldsymbol{D}_\theta^{(1)}, \ldots, \boldsymbol{D}_\theta^{(K)} \sim \mathcal{D}_\theta} \left[ \sum_{i=1}^n \min_j \mathbb{E}_{\boldsymbol{x} \sim \mathcal{U}(\Omega)} \left\| \hat{\boldsymbol{D}}_i(\boldsymbol{x}) - \boldsymbol{D}_\theta^{(j)}(\boldsymbol{x}) \right\|_2^2 \right], \tag{8}$$

where $\mathcal{U}(\Omega)$ is a uniform distribution over the scene bound $\Omega$. Eq. 8 only requires computing the pointwise difference between samples from $\hat{\mathcal{D}}(\boldsymbol{x})$ and $\mathcal{D}_\theta(\boldsymbol{x})$, making it efficiently optimizable with

---

[3]See the supplementary for the full derivation

gradient descent. Ultimately, **ProvNeRF** jointly updates the underlying NeRF's parameters and $\mathcal{D}_\theta$ by minimizing

$$\mathcal{L}_{\textbf{ProvNeRF}} = \mathcal{L}_{\text{NeRF}} + \mathcal{L}_{\text{fIMLE}} = \mathcal{L}_{\text{NeRF}} + \mathbb{E}_{\boldsymbol{D}_\theta^{(1)},\dots,\boldsymbol{D}_\theta^{(K)},\boldsymbol{x}} \left[ \min_j \mathbb{E}_{(\hat{t},\hat{\boldsymbol{d}})} \left\| (\hat{t},\hat{\boldsymbol{d}}) - (t_{j,\boldsymbol{x}}, \boldsymbol{d}_{j,\boldsymbol{x}}) \right\|_2^2 \right] \quad (9)$$

where $(t_{j,\boldsymbol{x}}, \boldsymbol{d}_{j,\boldsymbol{x}}) = \boldsymbol{D}_\theta^{(j)}(\boldsymbol{x})$, $(\hat{t}, \hat{\boldsymbol{d}})$ are i.i.d. samples from $\hat{\mathcal{D}}(\boldsymbol{x})$, and $\mathcal{L}_{\text{NeRF}}$ is the original objective of the NeRF model, e.g. photometric loss and depth loss. We provide implementation and architectural details in the supplementary material. See Figure 3 for the training pipeline illustration.

## 5 Experiments

Our ProvNeRF learns per-point provenance field $\mathcal{D}_\theta$ by optimizing $\mathcal{L}_{\text{ProvNeRF}}$ on a NeRF-based model. To validate ProvNeRF, we demonstrate that jointly optimizing the provenance distribution and NeRF representation can result in better scene reconstruction as shown in the task of novel view synthesis (Sec. 5.1). Moreover, we also show that the learned provenance distribution enables other downstream tasks such as estimating the uncertainty of the capturing field (Sec. 5.2). We provide an ablation study on fIMLE against other probabilistic methods in Sec. 5.3. Lastly, in Sec. 5.4, we show a preliminary extension of ProvNeRF to 3DGS [27].

**Stochastic Provenance Field Visualization** Fig. 7 visualizes the provenance stochastic field by sampling 16 provenances on a test view of the Scannet 758 scene. The directions of the samples are the *negative* of the predicted provenance directions for better illustration. Each sample is colored based on its predicted visibility. Notice that fIMLE allows ProvNeRF to predict multimodal provenance distributions at different scene locations.

### 5.1 Novel View Synthesis

We show modeling per-point provenance improves sparse, unconstrained novel view synthesis. As a point's provenances are sample locations from where the point is likely visible, the region between the provenance location samples and the query point should likely be empty. We design our provenance loss for NVS with this intuition.

Concretely, starting from a given NeRF model, we first sample points $\boldsymbol{x}_1, \dots, \boldsymbol{x}_N$ for a training camera ray parameterized as $\hat{\boldsymbol{r}}_x(t)$. Here we denote point $\boldsymbol{x}_i = \hat{\boldsymbol{r}}_x(\hat{t}_i)$. We only take points $\boldsymbol{x}_i$ with transmittance greater than a selected threshold $\lambda = 0.9$. For each visible point $\boldsymbol{x}_i$, we sample provenances $(t_1^{(i)}, \boldsymbol{d}_1^{(i)}), \dots, (t_K^{(i)}, \boldsymbol{d}_K^{(i)})$ from $\mathcal{D}_\theta(\boldsymbol{x}_i)$ with $\|\boldsymbol{d}_1^{(i)}\|_2 \geq 0.7$. Then each distance-

| | PSNR ($\uparrow$) | SSIM ($\uparrow$) | LPIPS ($\downarrow$) |
|---|---|---|---|
| Deterministic Field | 21.38 | 0.720 | 0.307 |
| Frustum Check | 21.56 | 0.728 | 0.297 |
| **Ours** | **21.73** | **0.733** | **0.291** |

Table 2: **NVS Ablation Results on Scannet.**

direction tuple $(t_j^{(i)}, \boldsymbol{d}_j^{(i)})$ gives a location $\boldsymbol{y}_j^{(i)} = \boldsymbol{x}_i - t_j^{(i)} \boldsymbol{d}_j^{(i)}$ from which $\boldsymbol{x}_i$ is observed. This in turn means $\boldsymbol{x}$ should be equally visible when rendered from ray parameterized as $\boldsymbol{r}_x^{(i)}(t) = \boldsymbol{y}_j^{(i)} + t\boldsymbol{d}_j^{(i)}, \forall j$. With this, we define our provenance loss for novel view synthesis as

$$\mathcal{L}_{\text{ProvNVS}} = \sum_{i=1}^{N} \sum_{j=1}^{K} \left[ \alpha + T(\boldsymbol{r}_x^{(i)}(t_j^{(i)})) - T(\hat{\boldsymbol{r}}_x(\hat{t}_i)) \right]_+, \quad (10)$$

where $[\dots]_+$ denotes the hinge loss and $\alpha = 0.05$ is a constant margin. $\mathcal{L}_{\text{ProvNVS}}$ encourages the transmittance at $\boldsymbol{x}_i$ along training camera rays to be *at least* the visibility predicted by the sampled provenances from the provenance field with margin $\alpha$. By matching transmittances between the provenance directions and the training rays, $\mathcal{L}_{\text{ProvNVS}}$ can be used together with $\mathcal{L}_{\textbf{ProvNeRF}}$ to optimize the NeRF representation and the provenance field, resulting in an improved scene geometry. We apply ProvNeRF to SCADE [57] for the task of novel view synthesis. See the supplement for details on the dataset, metrics, baselines, and implementation details.

**Results.** Table 1 shows our approach outperforms the state-of-the-art baselines in NVS on scenes from both the Scannet [13] and Tanks and Temples [29] dataset. Qualitative comparisons are shown

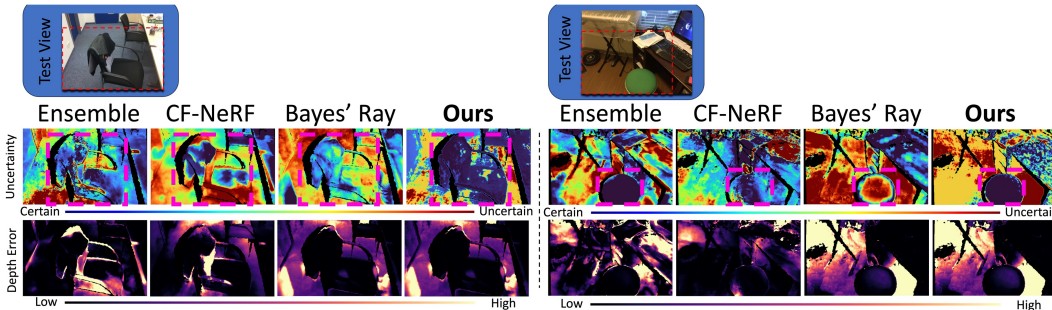

Figure 5: **Qualitative Results for Uncertainty Modeling.** We visualize our uncertainty maps obtained using the method described in Sec. 5.2. The uncertainty and depth error maps are shown with color bars specified. Uncertainty values and depth errors are normalized per test image for the result to be comparable. As shown in the boxed regions, our method predicts uncertainty regions with more correlation with the predicted depth errors.

| | Scannet | | | | Matterport | | | |
|---|---|---|---|---|---|---|---|---|
| | Avg. | #710 | #758 | #781 | Avg. | Room 0 | Room 1 | Room 2 |
| Ensemble | 7.71 | 3.01 | 2.96 | 17.2 | 63.0 | 8.04 | 110 | 71.3 |
| CF-NeRF [52] | 660 | 430 | 571 | 980 | 507 | 799 | 488 | 233 |
| Bayes' Rays [19] | 5.47 | 5.11 | 5.23 | 6.07 | 5.49 | 5.67 | 5.77 | 5.91 |
| **Ours** | **-3.05** | **0.19** | **-1.93** | **-7.40** | **-11.0** | **-13.6** | **-10.2** | **-9.17** |

Table 3: **NLL Results for Triangulation Uncertainty.**

in Fig. 4. We see that compared to the baseline SCADE, whose geometry is already relatively crisp, our $\mathcal{L}_{\text{ProvNVS}}$ can further improve its NVS quality by removing additional cloud artifacts, as shown in the encircled regions. Note that this improvement does not require any additional priors and is only based on the provenance of the scene.

We also compare our performance with deterministic baselines: *Deterministic Field* regresses one provenance for each 3D location using a neural network and *Frustum Check* calculates the training provenance defined in Eq. 4 by back-projecting the sampled points to one of the training camera and use that as the regularization information. Table 2 shows that our provenance field outperforms these baselines on the novel view synthesis task because the deterministic field cannot model complex provenance distribution and the frustum check baseline lacks generalization ability as it cannot be optimized to adapt the output provenance based on the current NeRF's geometry.

### 5.2 Modeling Uncertainty in the Capturing Process

Provenances allow for estimating the uncertainty in triangulation, i.e., the capturing process. In classical multiview geometry [20], the angle between the rays is a good rule of thumb that determines the accuracy of reconstruction. Fig. 6 illustrates this rule as the region of uncertainty changes depending on the setup of the cameras. Formally, for a 3D point $\boldsymbol{x}$, we sample provenances $\{(t_j, \boldsymbol{d}_j)\}_{j=1}^{K}$ from $\mathcal{D}_\theta(\boldsymbol{x})$. Treating $\boldsymbol{d}_j$ as the principal axes and $t_j$ as the distances from $\boldsymbol{x}$ to the camera origin, each provenance sample defines a pseudo camera $P_j$ that observes $\boldsymbol{x}$ at pixel location $x_j = \text{Proj}_j(\boldsymbol{x})$. Following chapter 12.6 of [20], we define the triangulation uncertainty of $\boldsymbol{x}$ as the probability of $\boldsymbol{x}$ given its noisy 2D pseudo observations: $\mathbb{P}(\boldsymbol{x}|x_1, \ldots, x_K) \propto \mathbb{P}(x_1, \ldots, x_K|\boldsymbol{x})\,\mathbb{P}(\boldsymbol{x}) = \prod_{j=1}^{K} \mathbb{P}(x_j|\boldsymbol{x})\,\mathbb{P}(\boldsymbol{x})$.

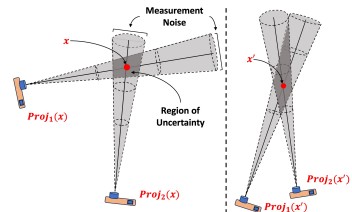

Figure 6: **Triangulation Uncertainty [20].** The figure shows that $\boldsymbol{x}'$ is more uncertain compared to $\boldsymbol{x}$ because the predicted provenances for $\boldsymbol{x}'$ give a narrower baseline than the baseline given by provenances of $\boldsymbol{x}$.

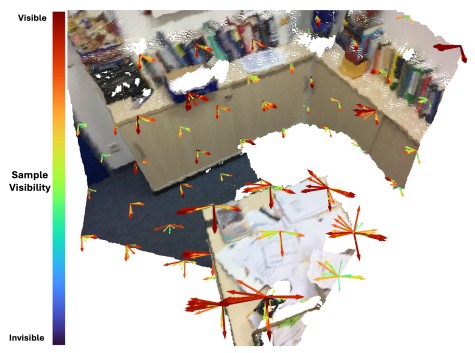

Figure 7: **Visualization of Provenance Field.**

| | AP ($\uparrow$) | AUC ($\uparrow$) |
|---|---|---|
| Deterministic Field | 0.163 | 0.168 |
| Gaussian-based w/ $C = 2$ | 0.537 | 0.539 |
| Gaussian-based w/ $C = 5$ | 0.629 | 0.631 |
| VAE-based | 0.323 | 0.325 |
| **ProvNeRF** w/ Spatial Inv. $\mathcal{Z}$ | 0.742 | 0.744 |
| **Ours** | **0.745** | **0.747** |

Table 4: **Ablation Results on Scannet.**

The last two equalities are derived by assuming independence of the 2D observations and each $\mathbb{P}(x_j|\boldsymbol{x})$ follows a Gaussian distribution $\mathcal{N}(\text{Proj}_j(\boldsymbol{x}), \sigma^2)$. This assumption is equivalent to corrupting each 2D observation $\text{Proj}_j(\boldsymbol{x})$ by a zero-mean Gaussian noise with $\sigma^2$ variance, accounting for measurement noises in the capturing process. Assuming a uniform prior of $\mathbb{P}(\boldsymbol{x})$ over the scene bound, the exact likelihood can be efficiently computed with importance sampling. This quantifies a point's triangulation quality given the sampled provenances, which becomes a measurement of the uncertainty of the capturing process. We apply our provenance field to ProvNeRF with different NeRF backbones [57, 49] and compute the likelihood. See supplementary for details on the dataset, metrics, baselines, and implementations.

**Results.** Tab. 3 shows the quantitative results on Scannet [13] and Matterport3D [9]. We follow [51, 55] to measure the negative log-likelihood (NLL) of the ground-truth surface under each model's uncertainty prediction. Since our **ProvNeRF** can be applied to any pre-trained NeRF module, we use pre-trained SCADE [57] for Scannet and DDP [49] for Matterport3D, both of which are state-of-the-art approaches in each dataset. Our approach achieves the best NLL across all scenes in both datasets by a margin because we compute a more fundamentally grounded uncertainty from classical multiview geometry [20] based on triangulation errors, while both CF-NeRF and Bayes' Rays require an approximation of the true posterior likelihood. Fig. 5 shows qualitative comparisons between baselines' and our method's uncertainty estimation. We expect a general correlation between uncertain regions with high-depth errors. An ideal uncertainty map should mark high-depth error regions with high uncertainty and vice versa. As shown in the boxed regions in the figure, our method's uncertainty map shows better correlation with the depth error maps. We also quantitatively evaluate uncertainty maps using negative log-likelihood following prior works Notice that in both examples, our method's certain (blue) regions mostly have low-depth errors (e.g., encircled parts in Fig. 5) because our formulation only assigns a region to be certain if it is well triangulated (Fig. 6). On the other hand, baselines struggle in these regions because they either use an empirical approximation from data or a Gaussian approximation of the ground truth posterior likelihood.

### 5.3 Ablation Study

We validate the choice of fIMLE as our probabilistic model by measuring the average precision (AP) [17] and area under the curve (AUC) [17] of predicted provenances $(t, \boldsymbol{d})$ against ground truth provenances $(\hat{t}, \hat{\boldsymbol{d}})$ for a set of densely sampled points in the scene bound. See supplementary for metric and ablation implementation details.

**Deterministic v.s. Stochastic Field.** We validate the importance of modeling per-point provenance as a stochastic field rather than a deterministic field. We model $\mathcal{D}_\theta$ with a deterministic field parameterized by a neural network. Table 4 shows the importance of modeling per-point provenance as a stochastic field. Since the provenances of a point are inherently multimodal, a deterministic field that only maps each $\boldsymbol{x}$ to a single provenance cannot capture this multimodality.

**Choice of Probabilistic Model.** We validate our choice of fIMLE [33] as our probabilistic model. We first compare with explicit probabilistic models that model the provenance field as a mixture of $C$ Gaussian processes and a VAE-based model. Table 4 shows results for the Gaussian Mixture field with $C = 2, 5$ and the VAE-based process. Although the performances for the Gaussian-based models improve as we increase $C$, they still suffer from expressivity because of their explicit density

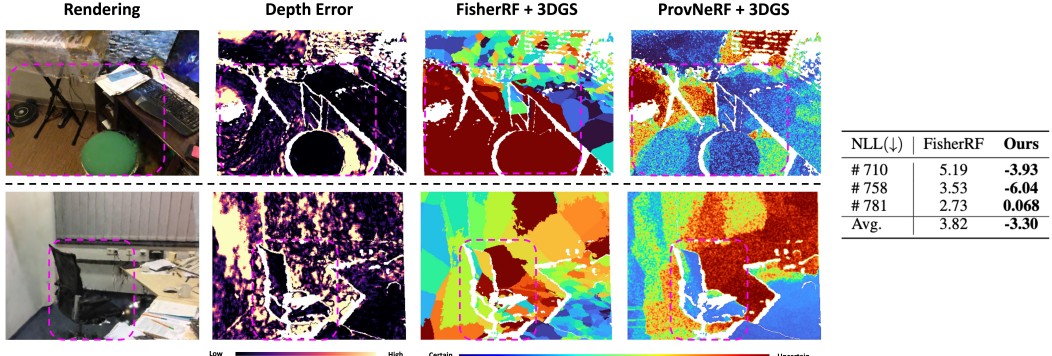

Figure 8: **Uncertainty Estimation Comparison with 3DGS.** Compared with FishRF, our method is able to estimate uncertainties that correlate more with the depth error as shown by the encircled regions. The right shows a quantitative comparison of uncertainty in negative log-likelihood.

assumption. Similarly, the VAE-based model suffers from mode-collapse while our fIMLE enables capturing a more complex distribution with a learned transformation $\boldsymbol{H}_\theta$.

**Choice of Random Function $\mathcal{Z}$.** Lastly, we validate our latent stochastic field $\mathcal{Z}$. We ablate our choice of $\mathcal{Z}$ with instead using a spatially *invariant* latent stochastic field $\mathcal{Z}^\star$ with $\mathcal{Z}^\star(\boldsymbol{x}) = [\boldsymbol{\varepsilon}, \boldsymbol{x}] \, \forall x$. Here, $\boldsymbol{\varepsilon}$ is a Gaussian noise vector in $\mathbb{R}^d$. Table 4 shows the comparison between $\mathcal{D}_\theta$ obtained by transforming $\mathcal{Z}$ (**Ours**) and transforming $\mathcal{Z}^\star$ (**Spatial Inv.** $\mathcal{Z}$). We see that using a spatially varying latent stochastic field further increases the expressivity of our model.

### 5.4 Preliminary Extension to 3D Gaussian Splatting

Because ProvNeRF is a post-hoc method that can model the provenance information for arbitrary novel view synthesis representations, we conduct a preliminary experiment that extends our provenance field modeling to 3D Gaussian Splatting [27]. Specifically, given a pre-trained Gaussian representation $\mathcal{G}$, we model a provenance distribution for each splat using IMLE with a shared 6-layer MLP for $\boldsymbol{H}_\theta$. The post-training takes around 30 minutes on a single A6000 Nvidia GPU. To show the usefulness of ProvNeRF applied to 3DGS, we use the methodology in Sec. 5.2 to estimate uncertainty maps and compare them with the predicted depth errors. Fig. 8 shows a qualitative comparison of our uncertainty map w.r.t. FishRF [25], a recent 3DGS uncertainty estimation baseline. Compared to their uncertainty map, ours shows more correlation to the depth error as highlighted by the boxed regions. Quantitatively we evaluate NLL on the three Scannet scenes shown on the right side of the same figure and show substantial improvements over FishRF. This improvement over existing literature suggests applying ProvNeRF to other representations such as 3DGS is promising. We leave further exploration of the method and applications as future works.

## 6 Conclusion, Limitation, & Future Works

We present ProvNeRF, a model that enhances the traditional NeRF representation by modeling provenance through an extension of IMLE for stochastic processes. ProvNeRF can be easily applied to any NeRF model to enrich its representation. We showcase the advantages of modeling per-point provenance in various downstream applications such as improving novel view synthesis and modeling the uncertainty of the capturing process.

We note that our work is not without limitations. Our ProvNeRF requires post-hoc optimization, which takes around 8 hours on SCADE, limiting its current usability for real-time or on-demand applications. However, the idea presented in our work is not specific to the model design and can be adapted to other representations. See Sec. 5.4 for preliminary adaption of ProvNeRF to 3DGS.

We also note that the hyperparameters to incorporate ProvNeRF are chosen for better performance, e.g. for the uncertainty and novel view synthesis applications, and in the future, it will be beneficial to explore a more adaptive approach in integrating provenance to different downstream applications.

**Acknowledgement** This work is supported by a Vannevar Bush Faculty Fellowship, ARL grant W911NF-21-2-0104, an Apple Scholars in AI/ML PhD Fellowship, and the Natural Sciences and Engineering Research Council of Canada (NSERC).

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
