# OpenReview forum: "ProvNeRF: Modeling per Point Provenance in NeRFs as a Stochastic Field"
_NeurIPS.cc/2024/Conference — NeurIPS 2024 poster_

### Official Review · Reviewer_o98v · 2024-07-10

**Soundness:** 3
**Presentation:** 2
**Contribution:** 2
**Rating:** 6
**Confidence:** 2

**Summary:**

Paper presents a method for co-training an inference network that learns where points in a NeRF were “seen from” (ie. provenance) in the training data. This allows for quantification of uncertainty of the reconstructed geometry in terms of triangulation (location) and depth error. Mathematically frames these these provenances within a stochastic process, extending implicit maximum likelyhood estimation (IMLE) to the functional domain over infinite (euclidian) sets. Presensts a loss term for incorporation into NeRF training and demonstrate superior novel view synthesis with it, and lower triangulation uncertainty on two standard datasets.

**Strengths:**

While I'm not convinced the "Methods" section is as crystal clear as it could be, and the problem domain widely applicable, the paper appears to present a novel contribution in the extension of IMLE to infinite sets and functional samples that should be of likely interest to the NeurIPS community.  The presentation is mathematically rigorous, though somewhat opaque, with extensive derivations in the supplement.

Demonstrates improvements to novel view synthesis of existing NeRF scenes through co-training with their additional loss term and  synthesis term, achieving improvements in reconstruction metrics. Also demonstrates quantitative SotA results over competing methods on triangulation uncertainty.

**Weaknesses:**

Introduction still leaves me asking “Why is this an important problem?”. The method only addresses triangulation uncertainty, ignoring other forms of uncertainty like transients, perhaps limiting applicability.

Overly mathematical presentation. Is this being presented in the absolute simplest way possible? Confusion over just exactly which the stochastic process is being discussed. Should clearly state the use-case (NeRF training add-on, or co-training objective?) up-front.

**Questions:**

The exposition of section 4.1 is unclear and took me a while to understand. Primarily, I understand that for each point in space $x$ there is a distribution of provenances, But I could imagine two stochastic processes: 1) where each point is sampled from the distribution for a fixed location, or 2) where each point becomes the provenance distribution for the next point drawn. This could be made clearer. Perhaps a diagram would help. (Also note having figure S1 in the main text would have gone a long well to help me understand the method better).

#25 "modeling for each point, the locations where it is likely visible.". Do you actually mean "likely visible in the training dataset", or "likely visible in the scene"? I believe you mean the former, yet this critical point was not clear to me until reading further into the paper. This seems like a critical distinction to make. ie. I could have a single view of a scene, yet properly infer that a point should be visible from many directions, even if not present in the training data.

#126 "forms a stochastic process, $\mathcal{D}_\theta$ indexed by coordinates $x \in \mathbb{R}^3$" Shouldn't this be more correctly defined as a "stochastic field", given it's indexing by Euclidean space. In general this is not the "stochastic process" most readers would usually be accustomed to, as one indexed by the natural positive numbers. Also, you are defining samples as functions, not as $\mathbb{R}^3$ positions, correct? Given this is the core of your contribution, extra clarity for readers would be appreciated for this.

#136, since you call $\mathbf{d}$ a "direction" it should be a normalized direction. Therefore the tuple should be in $R_+ \times \mathbb{S}^2$, right? ? If $\mathbf{d}$ is not normalized, I'm not sure I understand why (and should be called a "vector", not a "direction"). This should be a direction pointing towards the camera origin?

#138 I would define (5) before (4) for clarity. Also, presumably $\mathbf{o}_i$ is the camera origin? Though I don't believe it's been defined yet.

#211 Where is the actual architecture used for $\mathbf{H}_\theta$ described? This implementation detail seems critical to understanding the method, and how it might be helping NeRF training.

**Limitations:**

Authors leave discussion of limitations to a short paragraph in the supplemental. I’d prefer to see this in the main-text. Main current limitation is a long post-hoc training process. I’d also like to have seen a longer discussion on applicability of the method.

Authors sufficiently discuss the societal impact of their method.

---

> ### Author Rebuttal · Authors · 2024-08-07
>
> Thank you for finding our theoretical extension of IMLE to stochastic processes novel and mathematically rigorous and demonstrating improvements over existing methods. We address the questions raised below.
> ## Q1 Introduction still leaves me asking “Why is this an important problem?”
> Modeling provenances enables analysis of NeRF reconstruction from a classical stereo-matching point of view. As noted by the other reviewers, provenances are important concepts to model because they shed “insight about the quality of the input data for reconstruction” (GM1F) that “leads to a combination of traditional 3D vision-related fields, e.g. triangulation and uncertainty, to novel view synthesis” (u5T3). We will modify the introduction for better clarity and emphasis on the importance of provenance in the final version.
> ## Q2 Method only addresses triangulation uncertainty.
> Yes, one application of modeling provenances is modeling triangulation uncertainty in NeRFs. While uncertainty within NeRF’s reconstruction is multifaceted, isolating triangulation uncertainty can benefit downstream tasks such as next-view augmentation [19] that concern the capturing process (note other uncertainties such as transient are intrinsic to the scene). In the future, we hope to combine other types of uncertainty with triangulation uncertainty to achieve a more comprehensive understanding of NeRF reconstruction.
> ## Q3 Mathematical presentation; should clearly state the use case up-front.
> Thank you for your valuable feedback; we will modify our paper accordingly in our final version. Our ProvNeRF is a NeRF training add-on and we will state that up-front. We intend to describe the method in a mathematically precise way, but instead we will in the final version lighten the mathematical notations, simplify the idea's presentation and move details to the supplementary.
> ## Q4 Stochastic process in the exposition of Sec 4.1.
> Our stochastic process is defined as a collection of distributions at each 3D location instead of distribution over different 3D locations. To provide an illustration, Figure 2(a) in our rebuttal shows the provenance direction samples at different 3D locations. We will include this in the final version and add further clarifications to avoid confusion.
> ## Q5 Clarification on visibility.
> By visibility, we refer to that given by the training dataset (so the former in your question), not from the scene. We will clarify this in our final version. But modeling visibility of the scene would be an interesting follow-up work.
> ## Q6 Stochastic process v.s. Stochastic field.
> Yes, stochastic field would be a more suitable name for our provenance model, as each of our samples is a function mapping each 3D location to a provenance of it. We note that there is an inconsistency in the stochastic process literature in terms of terminology. For example, [74] defines stochastic process as an n-dimensional and m-variate random function whereas [75] defines it as those with only a one-dimensional indexing set. However, stochastic field is a more suitable name given that our indexing set is a field. We will change the stochastic process to the stochastic field in our final version. Note that we still use the term provenance stochastic process in rebuttal for terminology continuity.
> ## Q7 Why are directions not unit length?
> The predicted direction samples are not unit length to handle object/scene occlusions. We model a visibility term $v \in [0,1]$ (c.f., Eq. (5), main) as the norm of the predicted direction that accounts for how occluded this provenance sample is: this is why the distance-direction tuple lives in $\mathbb{R}_+ \times \mathbb{D}^3$. We overloaded the notation and used $\bf d$ for both the unnormalized and normalized directions. In practice, once we extract a provenance’s visibility by computing the norm of the direction, we normalize the direction and use it for downstream tasks. In the final version, we will denote the normalized direction as $\tilde{\bf d}$ and fix the inconsistencies in the paper. For further clarity, we note that our visualizations show the negative of the provenance direction (for clearer visuals) – that is the direction from a 3D location should be pointing away from the camera.
> ## Q8 $o_i$; defining Eq. (5) before (4).
> Yes, $o_i$ is the camera origin (see Ln. 99 main). We will reorder Eq. (4) and (5) in the final version; thank you for your suggestion.
> ## Q9 Architecture of $H_\theta$.
> It is a 3-layer MLP with ReLU non-linearity and input feature, hidden feature, and output dimensions being 288, 256, and 4, respectively. We will include these details in Sec. S2 in the final version.
> ## Q10 Limitation.
> Thanks for the suggestion. We will move our limitation discussion to the main paper.
> ## Q11 Longer discussion on method’s applicability.
> Our method can be applied to model stochastic fields other than provenance fields. For example, an interesting application other than provenance modeling is to model the material properties of a NeRF as a stochastic field. For instance, we could model the BRDF at each point in 3D as a stochastic field over the incoming and outgoing solid angles. Learning such a stochastic field would allow us to enable interesting applications such as relighting or material modification. Another possibility is to extend the deterministic framework in [76] to model the equivalent classes of signed distance fields given by a set of discrete signed distance samples as a stochastic field. This allows us to sample different but all plausible signed distance fields (thus implicit surfaces) given a set of discrete signed distance samples. We will include this discussion in the final version.
>
> [74] Shinozuka, M. (1987). Stochastic Fields and their Digital Simulation.
>
> [75] Knill, O. (2009) Probability Theory and Stochastic Processes with Applications.
>
> [76] Sellán, S., et. al. (2024). Reach For the Arcs, SIGGRAPH 24’

---

> > ### Comment · Reviewer_o98v · 2024-08-09
> >
> > I thank the authors for their thorough rebuttal.
> >
> > Addressing the points as you have outlined will go a long way towards making this an excellent paper. As the practical application and evaluation metrics are somewhat weak, I do think the theoretical insight holds promise. Making the theoretical presentation crystal clear will make this paper a valuable read for the research community.
> >
> > Having more compelling use cases and conclusive evaluation metrics would push me into "accept" or "strong accept" territory.

---

> > > ### Author Response · Authors · 2024-08-11
> > >
> > > Thank you for taking the time to review our paper and rebuttal, and for recognizing the theoretical promise of our work. We promise to revise the notations and rearrange the presentation to clarify the theoretical analysis in the final version.
> > >
> > > While our improvements in Table 1 against baselines are modest, it’s important to note that our method is different from the baselines because we don’t introduce additional priors into the optimization process. Essentially, the geometry improvements showcased in Figure 3 come at no extra cost, thanks to our modeling of provenances. This characteristic also makes our method easily integrated into other approaches to further improve reconstruction. When compared with Bayes’ Rays [19], who, like us, can be plugged into any NeRF representations to improve its reconstruction without using additional priors, our method performs significantly better. The following shows NVS metrics comparison on the Scannet dataset between Bayes’ Rays and ours using the same pretrained SCADE model:
> > >
> > > |Scannet|Average|Scene 710|Scene 758|Scene 781|
> > > |:-:|:-:|:-:|:-:|:-:|
> > > ||PSNR/SSIM/LPIPS|PSNR/SSIM/LPIPS|PSNR/SSIM/LPIPS|PSNR/SSIM/LPIPS|
> > > | Bayes’ Rays |20.09/0.707/0.304|21.05/0.692/0.335|18.75/0.741/0.284|20.46/0.688/0.294|
> > > | **Ours** |**21.73**/**0.733**/**0.291**|**21.48**/**0.703**/**0.328**|**21.99**/**0.786**/**0.258**|**21.70**/**0.711**/**0.288**|
> > >
> > > While we cannot post rendering comparisons, the metrics above indicate that Bayes’ Rays’ regularization actually significantly degrades SCADE’s NVS quality because it removes all the uncertain regions as a post hoc operation – this operation in practice removes actual geometry rather than floaters and thus causes the degradation. In comparison, our formulation removes most of the visual artifacts as shown in Figure 3 of main and Figure 2 (b) in the rebuttal PDF without affecting the scene reconstruction. This is because our provenance stochastic process allows us to formulate an entirely differentiable regularizer and thus can improve the scene geometry through co-training instead of as a post-hoc operation.
> > >
> > > In addition to the applications in our experiment section (Sec. 5), we also provide an additional application in Sec. S5 that uses provenance to select favorable camera viewpoints based on differentiable criteria, leveraging a neural rendering framework. For instance, this can encompass orienting the camera to align with the normal vector of a specified target or achieving a detailed close-up view of the target. By incorporating provenances into the optimization objective that we define in Sec. S5, we are able to obtain a camera viewpoint that satisfies the predefined objective while achieving good rendering quality as shown in Figure S6 where we compare our formulation with a retrieval-based and a provenance-agnostic optimization-based baseline.
> > >
> > > We hope we have addressed your concerns and questions. Please do not hesitate to ask us should there be anything unclear. Lastly, we hope you can take the use cases and comparison delineated above into consideration for your final decision.

---

> > > > ### Comment · Reviewer_o98v · 2024-08-12
> > > >
> > > > I thank the authors for their response. I still maintain my rating as "Weak Accept". I appreciate the points you have made, but still fall short of the "compelling use case" or "conclusive metrics" I would require for a stronger rating.

---

### Official Review · Reviewer_GM1F · 2024-07-13

**Soundness:** 3
**Presentation:** 2
**Contribution:** 3
**Rating:** 6
**Confidence:** 3

**Summary:**

The paper describes a method for explicitly modeling the visibility of a point in space in a NeRF model. This is done though modeling provenance which is the space of points where the given point is likely visible. The motivation is that modeling visibility enables the underlying NeRF model to better utilize triangulation information from the training images. The proposed method introduces a neural network to model the provenance function which allows for sampling points visible point estimates from a given 3D position. This secondary network can be applied to any base NeRF model. When trained with the provenance loss, the resulting model performs better in the task of novel view synthesis across a variety of scenes and metrics, and allows for explicit modeling of uncertainty in the reconstructed scene geometry.

**Strengths:**

+ The motivation for the problem is interesting. If you have a way to approximate where a point is visible given the input data, that does provide a lot of insight about the quality of the input data for reconstruction of view synthesis.

+ The results shown are good. The overall metric while better than other approaches are just a slight bump, but there are several examples shown where some of the common artifacts present in indoor NeRFs are mitigated if not completely removed using this method. That improved visual quality on top of the benefits of uncertainty modeling make this quite compelling.

+ The ablation in Table 2 shows that the naive ideas that most people would have tried are worse than this, so I think the more obvious baselines are covered in comparison. Similarly Table 4 is a good ablation over several methods that I think would be the go to baseline for an idea like this. Overall the evaluation is very thorough.

+ The main paper + supplemental is an immense amount of information and evaluation for a single paper. Clearly a lot of effort was put into describing and validating the method.

**Weaknesses:**

- Overall I found the presentation of the method to be very confusing. There is a lot of technical overview in 4.1-4.3 that I think over complicates what is actually happening. I had to reread these sections several times while making passes through the full paper to really get a grip on what is happening. The architecture diagram in Figure S1 makes it much clearer. I think the extremely technical discussion could be reduced and moved to the supplemental while a more direct description of what is actually being done like in the supplemental should be focused on more.

- I don't follow why norm(d)>=0.7 has any berrying on the confidence of that provenance sample. The output provenance sample D(x) = (t,d) where the point y where x is estimated to be visible is y=x-td. So why is the length of the direction vector a signal for uncertainty?

- The visualizations of the provenances for points don't align with what I expect. Why are the vectors all roughly the same length? Shouldn't it be more varied? Similarly, I don't think the definition of how to recover the point y where x is visible is correct based on the description and visualization. If y = x-td, then the output direction vector would be pointing away from points of visibility, not towards.

- To reiterate my first point, the major weakness of this paper is the somewhat overwhelming amount of technical detail for what boils down to a pretty simple idea. Maybe if the details are reordered so that the simple explanation comes first followed by the more rigorous definition the paper would be easier to follow. After several reads I can follow the early sections, but they are still not extremely clear. I would recommend simplification if possible to make the core contribution as clear as possible and fill in the details where necessary instead of the current presentation. Though I admit that the technical groundwork is necessary so it is a difficult balance. But a clear example of what can be simplified is section 4.3. There is the initial equation 7 discussion, followed by the logic for simplifying to equation 8, which eventually reduces to equation 9 which is what I think most people would have assumed would be the natural conclusion. It's not to say that this presentation of the principles from equation 7 to equation 9 is not important if useful, but it is a bit heavy handed in my opinion.

**Questions:**

End of line 128,  "we" should be capitalized.

This is not necessarily a weakness, but it would be good to clarify. Are the 3D models being shown in the visualization based on evaluating the provenance on the ground truth mesh from depth fusion, or is that somehow extracted from the NeRF itself. They are shockingly clean to be extracted from a NeRF but it's not clear to me from the text if that is actually what's happening.

Line 168 " We define a latent function sample Z ~ Z to be the concatenation of a random linear transformation of x and x itself." I don't think concatenation is the right word here based on equation 6. I think this sentence is phrased poorly or I'm missing something important.

**Limitations:**

Yes

---

> ### Author Rebuttal · Authors · 2024-08-07
>
> Thank you for finding our work interesting and insightful and our evaluations thorough and compelling. We address the questions raised below.
> ## Q1 Presentation of the method section.
> Thank you for your valuable feedback. We will incorporate your suggestions and promise to improve the presentation of the method in the final version. We originally intended to describe the method in a mathematically precise way, but we understand this can cause confusion. We will lighten the mathematical notations, simplify the presentation of the idea, and move details to the supplementary. We will describe the method in a more intuitive way in the final version.
> ## Q2 Clarification on length and norm of direction vector.
> The length of the predicted direction determines how visible the provenance direction is to handle object/scene occlusion. That is, we model a visibility term $v \in [0,1]$ (c.f., Eq. (5) of main) as the norm of the predicted direction that accounts for how “visible” this provenance sample is. For further clarity, we note that we overloaded the notation and used $\bf{d}$ for both the unnormalized and normalized directions. In practice, once we extract a provenance’s visibility by computing the norm of the direction, we normalize the direction and use it for downstream tasks. In the final version, we will denote the normalized direction as $\tilde{\bf d}$ and fix the inconsistencies in the paper. Finally, in our NVS regularizer formulation in Eq. (10), we use the length of the direction vector to distinguish floaters from solid surfaces. I.e., if the predicted direction has $\lVert\bf d\lVert<0.7$, we assume that there is a solid surface in that direction and it shouldn't be regularized. Otherwise if $\lVert\bf d\lVert\geq0.7$, we treat it as a floater where we apply our regularizer to remove it. We will add this clarification in the final version.
> ## Q3 Variation in length of direction vectors.
> They are varied, as shown by the length of vectors in the supplementary video. We further provide additional visualization in Figure 2 (a) in the rebuttal PDF; we will include this in the final version. We note that the visualization in Figure 6 only shows normalized directions. We apologize for this confusion and we will change it to reflect the length of the direction vector in the final version.
> ## Q4 Misalignment of provenance visualization.
> Yes, you are right. The visualizations in Figure 6 and the video show the negative of the predicted provenance directions. This is done for ease of illustration: otherwise the samples will be hidden by the scene. We will add this clarification in the final version.
> ## Q5 Ln. 128.
> Thanks for pointing this out. We will fix it in the final version.
> ## Q6 Clarification of the provenance visualization.
> Yes, the visualizations shown in the video and Figure 6 are provenance samples generated at the ground truth geometry surface provided by the dataset. We will include this clarification in the final version.
> ## Q7 Ln 168.
> Thanks for the suggestion. Yes, the expression in Eq.(6) is a more mathematically precise definition of our random function $\bf{Z}$. We will modify the wording for a more precise definition in the final version.

---

### Official Review · Reviewer_u5T3 · 2024-07-13

**Soundness:** 2
**Presentation:** 3
**Contribution:** 3
**Rating:** 5
**Confidence:** 4

**Summary:**

This paper addresses gaps in existing Neural Radiance Fields research by modeling the provenance of each point as a stochastic process and enhancing triangulation quality through an extended Implicit Maximum Likelihood Estimation (IMLE) to functional space, resulting in improved novel view synthesis and uncertainty estimation under sparse, unconstrained view conditions. Experimental results demonstrate the effectiveness of the proposed method.

**Strengths:**

1. The idea is interesting, especially the defined problem in this paper makes a lot of sense to me. ProvNeRF enhances Neural Radiance Fields by integrating per-point provenance information during training, which enriches the model with critical insights on triangulation quality. This integration leads to a combination of traditional 3D vision-related fields, e.g. triangulation and uncertainty, to novel view synthesis, particularly under challenging sparse and unconstrained viewing conditions.
2. The writing is very clear and easy to follow.

**Weaknesses:**

1. My main concern is about the evaluation of the proposed idea. According to the paper, modeling the provenance for each point could potentially improve NeRF's accuracy in predicting the positions of points in space. However, there are few direct comparative experiments on depth/position, with only Fig. 4 showing some reflection. The depth error does not seem to significantly differ from other methods. This makes the evaluation less convincing from my view.
2. From Table 1,2, I wouldn't say that the result is significantly improved. And, the visualization in Fig.4, the difference between the proposed one and Bayes' is not notable. These make me a little bit concerned about the significance of the proposed method as it requires an 8-hour post-optimization.

minor
1. The description of the images in Fig. 2 is reversed, with left and right sides swapped.

**Questions:**

1. Does the proposed method solve the uncertainty problem? Does the uncertainty problem really matter in the NeRF-based method? If so, why does the proposed method have inferior performance compared with other sOTA methods in evaluation, e.g. table 1?
2. Can this method be applied to 3DGS or INGP-based methods? How much time it will cost to add the proposed framework in the post-optimiazation?

**Limitations:**

The authors demonstrate shortcomings concerning the optimization duration and mention a solution to address these deficiencies in the appendix section.

---

> ### Author Rebuttal · Authors · 2024-08-07
>
> Thank you for finding our method interesting, easy to follow, and enriches NeRFs with critical insights on triangulation quality.
>
> We here clarify our experiment setup and avoid confusion. Our main experiments are split into two parts, each corresponding to a different application using ProvNeRF. 1) Novel view synthesis (Sec. 5.1): we use our learned provenance to formulate an additional regularizer $\mathcal{L}\_{provNVS}$ to improve scene reconstruction. We evaluate our approach in Fig. 3 and Tab. 1, demonstrating improvements compared to baselines. 2) Triangulation uncertainty estimation (Sec. 5.2): we use our learned provenance to estimate triangulation uncertainty. We show results on uncertainty estimation in Fig. 4 and Tab. 3 against baselines by measuring the uncertainty’s correlation to depth errors. This is a common evaluation protocol in prior works [51-52]. Note that in this context we do not compare the depth error across different models. We address your questions below.
> ## Q1 Evaluation of the proposed idea.
> As highlighted by other reviewers, we provide a “thorough” (GM1F) evaluation of our method and achieve improvements in both reconstruction and triangulation uncertainty metrics(o98v). In the attached PDF, we provide additional experimental results, including applying ProvNeRF on 3DGS for uncertainty estimation (Figure 1) and a depth map comparison between our method and SCADE (Fig. 2(b)).
> ## Q2 “Modeling the provenance improves NeRF's accuracy”.
> We clarify that modeling the provenance alone does not improve the NeRF’s reconstruction. However, it can be used to formulate the $\mathcal{L}\_{provNVS}$ regularizer to improve NVS and scene reconstruction as shown by experiments in Sec. 5.1. We further provide improved depth rendering visuals in Figure 2(b) of the attached PDF that supplements Fig. 3.
> ## Q3 “Few direct comparative experiments on depth/position, with only Fig. 4 showing some reflection.”
> We clarify that the depth error maps in Figure 4 are NOT to show improvement in depth as it is part of our uncertainty experiments (Sec. 5.2). Note that our $\mathcal{L}\_{provNVS}$ regularizer is not used here. Instead, Fig. 4 visualizes uncertainty maps from different methods and compares their correlations to the depth errors, which serve as a qualitative comparison for uncertainty estimation.
> ## Q4 Tables 1 and 2: “I wouldn’t say results significantly improved”.
> As noted by GM1F, while the improvements in the quantitative metrics on the NVS experiments (Table 1 & 2) are not huge, “several examples where some of the common artifacts present in indoor NeRFs are mitigated if not completely removed.” making our method “quite compelling”. This suggests that our additional regularizer $\mathcal{L}\_{provNVS}$ can improve the NeRF reconstruction by modeling provenance that comes for free from the training images.
> ## Q5 Difference between Bayes’ Rays in Figure 4.
> Sec. 5.2, Fig. 4 compares our approach against baselines such as Bayes’ Rays on uncertainty estimation. Note that a good uncertainty map should correlate well with depth error maps, marking regions with high depth errors as highly uncertain and vice versa. The boxed regions in Fig. 4 demonstrate the superiority of our estimated uncertainty over Bayes’ Rays. For example, our method accurately identifies both chairs (left and right examples) as certain due to good triangulation, contrasting with baselines that incorrectly mark them as uncertain despite their low depth errors. Again, we clarify that the results in Fig. 4 (main) do not show improvement in scene reconstruction – we do not enforce our $\mathcal{L}\_{provNVS}$ regularizer here. In fact, the depth error for Bayes’ Rays and ours are the same in Fig. 4 as we use the same base model to compute for uncertainty. For improvements in scene reconstruction see our NVS experiments (Sec 5.1.) where we enforce the regularizer to clear out floaters.
> ## Q6 Figure 2 caption.
> Thanks for pointing this out. We will fix this in our final version.
> ## Q7 Does the method solve the uncertainty problem and why is it important?
> Yes, our proposed method can be used to estimate triangulation uncertainty as shown in Sec. 5.2. Uncertainty estimation is crucial for applications needing reliable reconstruction like continual learning and robotic navigation. Several studies [51-52, 55] have addressed this within NeRF settings.
> ## Q8 Results are not SotA.
> We achieve SotA results in both NVS and uncertainty estimation applications. In the NVS experiment, Figure 3 shows our results clearly improve the previous SotA method SCADE by removing floaters and fuzziness on surfaces. This is also reflected quantitatively in Table 1 as we achieve on average better than the baselines on both datasets and ours are better in almost all the metrics; In the uncertainty experiment, qualitative and quantitative comparisons suggest that our uncertainty map is more informative -- i.e., marking poorly triangulated regions with high uncertainty and vice versa -- compared with existing baselines.
> ## Q9 Applying ProvNeRF to 3DGS.
> Yes, we can apply our method to 3DGS. We apply ProvNeRF on top of 3DGS on Scannet dataset. The post-optimization takes around 30 minutes per scene on a single NVIDIA A6000 GPU. To study its applicability, we used our trained provenance stochastic process to estimate triangulation uncertainty as in Sec. 5.2. See our global response for implementation details. Figure 1 in the attached PDF shows the comparison of estimated uncertainty maps between ours and FisherRF [73]. Note that we obtain a more correlated uncertainty map w.r.t. the estimated depth error from 3DGS. The same figure also shows a NLL comparison against FisherRF where we outperform the baseline by a large margin. These results show that applying ProvNeRF to recent explicit representations such as 3DGS is promising and we leave further exploration as future works.
>
> [73] Jiang, W. et. al. (2023). FisherRF, CVPR 24’

---

> > ### Comment · Reviewer_u5T3 · 2024-08-13
> >
> > I am grateful for the author's reply. The rebuttal partially addressed my concerns, i.e. applying the proposed method to 3DGS. However, I am still unconvinced with the evaluation. Feeling the same with GKL5, it is weird to connect uncertainty directly with depth error. I checked with [51-52]. Although they use uncertainty in the evaluation, depth map, predicted depth, and depth errors are also shown in the figures to demonstrate their superiority. This makes the presentation of the evaluation confusing and has a large space to improve.
> >
> > Moreover, as the depth map shown in the pdf, the proposed method has a better estimation in the edge region of the image compared to SCADE, i.e. some guess estimation vs blank, and remains the similar quality in the middle region compared to SCADE, i.e. see the USA flag in the second column in pdf.fig.3. It seems the proposed method is good at filling the blank, not improve the reconstruction quality.
> >
> > Overall, I understand the evaluation is okay in terms of uncertainty. However, there is still a gap between the motivation and the evaluation in terms of reconstruction quality. Thus, I will maintain my original rating score.

---

> ### Author Response · Authors · 2024-08-13
>
> We thank the reviewer for reading our rebuttal and eliciting their concerns which we address below.
> ## Weird to connect uncertainty directly with depth error
> We follow prior works [19, 51-52, 73] to visually evaluate our uncertainty estimation by inspecting its correlation with the depth error in Figure 4 of the main. We didn’t put the rendered depth maps in the figure because **we are a post-hoc uncertainty estimation method**, i.e. our approach is a plug-in to any NeRF backbone to estimate triangulation uncertainty, which means that **the depth maps we obtain will be exactly the same as the backbone**. However, baselines like CF-NeRF [51] and Stochastic NeRF [52] are not post-hoc methods and need to be trained from scratch with a modified volumetric rendering pipeline. So their convergence is not guaranteed and in fact, produces blurrier results than SCADE, the backbone we use. Our work is more similar to the recent baseline Bayes’ Rays [19], which also only shows a comparison between the uncertainty map and the depth error map and does not show rendered images or depths of the backbone (c.f. Fig. 6 of [19]).
>
> ## Comparison with SCADE; the USA flag in the second column in pdf. Fig.3.
> The error in the USA flag in Fig. 3 of the main is due to a pose inaccuracy on the dataset released by SCADE, a common issue from using COLMAP to estimate camera poses. This causes inconsistencies in the optimization, resulting in the wrong geometry of the flag. In fact, all of the methods in Table 1 take these poses as input and they all have this problem as well. This can be mitigated if we obtain better camera poses for the dataset. However, **it is important to note that this problem is orthogonal to our contribution in Sec. 5.1 where we use our provenance stochastic process to remove artificial floaters**, which is a common artifact in indoor NeRFs as suggested by GM1F and is mitigated by our method as shown in Fig. 3.
>
> ## It seems the proposed method is good at filling the blank, not improve the reconstruction quality.
> **Our method does not fill in the blank**. We are an optimization-based method and do not use any generative priors. Instead, **the improvements in Fig. 3 are direct results of our method removing floaters from the pretrained SCADE model and revealing the correct geometry behind them**. This is also suggested by the depth maps renderings we show in Fig 2 (b) of the rebuttal PDF.
>
> We hope we have answered your questions and concerns so that they can assist your final decision. Please do not hesitate to ask us should there be anything unclear.

---

### Official Review · Reviewer_GKL5 · 2024-07-18

**Soundness:** 3
**Presentation:** 2
**Contribution:** 3
**Rating:** 5
**Confidence:** 3

**Summary:**

This paper introduces a way to jointly learn/model provenance during NeRF training, where provenance is defined as locations where a 3D point is likely visible. This design is motivated by the classic idea of modeling triangulation quality, To implement this in NeRF, this paper extends implicit maximum likelihood estimation (IMLE) to functional space with an optimizable objective. Modeling provenance is beneficial to NeRF final results, and also enables a new way of uncertainty estimation.

**Strengths:**

- The introduced idea of Provenance modeling is sound and motivated by classic idea in stereo matching.
- The ability of estimating uncertainty is very important, and it's a free side-product of the introduced Provenance modeling.
- The idea of using Implicit Maximum Likelihood Estimation in functional space is indeed a good solution for represents probabilistic distribution as a set of samples, and also for stochastic processes.
- This paper also builds up a system of theory which is complete and reasonable. This theoretical framework will inspire many follow-up work in this field.
- The appendix is very informative and discussed things like extra results, important alternative designs, key derivations and ablation studies. I found it to be a good complement of the main paper.
- The findings of this method is helpful to sparse, unconstrained view setting is very encouraging.

**Weaknesses:**

I find a few things can be improved as listed below:
- Some motivations are less clear or need clarification, specifically:
  - L36 "we propose to model the provenance as the samples from a probability distribution, where a location y is assigned with a large likelihood if and only if x is likely to be visible from y": What's the alternative ways of modeling this probability distribution rather than using samples? Is it possible to learn a continuous representation rather than use discrete samples?
  - For sample-based generative model, what are the alternatives to implicit maximum likelihood estimation (IMLE)? Why is this specific approach is taken? A discussion could be very useful here.

- The visual results are hard to interpret. Denser and more informative caption can be considered.
  - Why depth error is always shown with uncertainty. What's the connection and how can we interpret the depth?
  - Compared to the baselines such as Bayes' Rays, what are considered as improvement from the uncertainty map? Is it more semantic? Or is it more correlated with depth, means near objects tends to have lower uncertainty.
  - From results it appears some floaters/fuzziness are addressed but not totally. I'm wondering what's the reason and how can we address the remaining artifacts.

- Since there is no GT for uncertainty, what's the best way to evaluate it? In Tab.3 NLL is used, is this the best way?

- L135, provenances is parameterized as a distance-direction tuple. What's the other options here? Can we directly model it as a 3D vector? Or using other parameterization of 3D vector?

- Sec.2 first paragraph "NeRFs and their Extensions" missed dicussion of recent explicit representation such as 3DGS. Also, it's unclear whether the proposed solution can generalize to 3DGS? In theory it should work but we don't know in reality. This extra backbone experiment can make the results and theory more convincing.

- This paper claims the proposed method helps  sparse, unconstrained view setting. Following this statement, the experiments should use a sparse setting such as few-shot NeRF. Also, I'm wondering why scannet and tanks and temple are picked as the testing benchmark but not other popular NeRF dataset (indoor, outdoor, synthetic, object-centric, etc).

**Questions:**

After reading, couple of questions remain:
- In Fig.2, how does camera baseline distance, occlusions/visibility, and stereo range errors relate to Provenance? The connection is less clear.
- Is it possible to visualize the learned confident provenances together with the camera poses in a way that can better illustrate the correctness/soundness of the learned provenances?
- For results in Tab.1, is `ours` built upon NeRF? If so, is the only difference between `ours` and `NeRF` is the extra loss?
- In Eq.9 there are two expectations. So in practice, two samplings are needed. I'm wondering how are D, x, and (t,d) sampled in practice.
- How important are L_provNVS and L_provNeRF individually? Which one is more important and why both are needed?

**Limitations:**

Limitations are discussed in the appendix where the authors are upfront about one important limitation of the running time.
Societal impact is also discussed in appendix.

---

> ### Author Rebuttal · Authors · 2024-08-07
>
> Thank you for finding our method sound, theoretically complete, and has the potential to inspire follow-up works in the field. We answer the questions raised below.
> ## Q1 Alternatives in modeling the probability distribution.
> Instead of as samples, we can either represent provenances as a discrete distribution in Eq.(4-5) main or as a closed-form continuous distribution (e.g. Gaussian Mixtures). However, discrete representations incur discretization errors and closed-form distributions have limited expressivity. In contrast, our provenances use an expressive deep net $H_\theta$. See Table S1 for an NVS quality comparison.
> ## Q2 Alternatives to functional IMLE.
> Compared to other sample-based implicit probabilistic models like GAN and VAE which are prone to mode collapses, IMLE can model multimodal distributions by implicitly maximizing likelihoods. Table 4 and Table S1 show quantitative comparisons of fIMLE against VAE highlighting its superior performance. We also tried GAN but it didn’t converge.
> ## Q3 Visual results and caption improvement.
> We include additional visualizations in the rebuttal. In Fig. 2 (a), we show provenances sampled on the ground truth surface. In Fig. 2 (b), we show improvements to depth images after applying $\mathcal{L}\_{provNVS}$ to SCADE. We will include them in the final version and update captions for Fig. 3, 4, and 6 of main for clarity.
> ## Q4 Uncertainty’s connection with depth error.
> A poorly triangulated region likely has incorrect depth due to large depth ambiguity (c.f. Fig. 5). Consequently, regions with high depth error should have high triangulation uncertainty and vice versa. Thus, we validate our uncertainty map by examining its correlation with depth errors, which is a common approach in prior works [51-52].
> ## Q5 What are considered improvements in uncertainty maps?
> A good uncertainty map should correlate well with depth errors. E.g., the boxed regions in Fig. 4 (main) demonstrate the superiority of our uncertainty over the baselines. Our method accurately identifies both chairs as certain due to low depth errors while baselines incorrectly mark them as uncertain.
> ## Q6 Floaters/Fuzziness not addressed totally.
> Fig. 3 (main) shows visuals from the NVS experiment using our $\mathcal{L}\_{provNVS}$, improving scene reconstruction by clearing most floaters. Rendered depths included in Fig. 2 (b) of the rebuttal show further validation. While our method in theory may not eliminate all the artifacts when they are invisible from training cameras, in practice it removes most floaters as shown in the figures above. In Fig. 4 (main) that compares uncertainty estimation methods, floaters are present because we didn't apply the $\mathcal{L}\_{provNVS}$ regularizer; these floaters are from the pretrained SCADE.
> ## Q7 Uncertainty evaluation.
> NLL, a common uncertainty metric [51-52, 55], measures the negative log-likelihood of ground truth surfaces under the model's uncertainty prediction. The metric is intuitive as effective uncertainty maps should assign high likelihood to the true scene surface. While Area Under Sparsification Error (AUSE) is also used [19. 73], Figure S5 shows its defects that can lead to unreliable scores. We therefore opted to evaluate NLL.
> ## Q8 Provenance parameterization.
> Yes, we can model provenance as a 3D vector that connects the observation center with the 3D location. Other parameterizations such as a camera pose in se(3) are also viable.
> ## Q9 3DGS discussion.
> Our method is compatible with 3DGS. We integrate ProvNeRF to 3DGS and assess its triangulation uncertainty using the method in Sec. 5.2. See Fig. 1 of the attached PDF for both visual and quantitative comparisons with FishRF [73]. The results demonstrate the potential of applying ProvNeRF to explicit representations like 3DGS, and we plan to explore this further in the future. Details are in our global response.
> ## Q10 Experiment setting: should be sparse and why use scannet, tanks and template dataset.
> Our experiments indeed use a sparse setting following SCADE [58]. Specifically, ScanNet and T&T are standard datasets [58] with 18-26 training images in unconstrained camera poses in each scene. We also test NVS on the In-the-Wild dataset from [58] achieving superior metrics over SCADE:
> ||PSNR|SSIM|LPIPS|
> |-|-|-|-|
> |SCADE|22.82|**0.743**|0.347|
> |Ours|**22.85**|**0.743**|**0.343**|
> ## Q11 Fig. 2 clarification.
> Fig. 2 illustrates that modeling provenance is not straightforward, as different triangulation-related phenomena [20] such as camera baseline distance and occlusions need to be taken into account. We’ll improve this figure and caption in the final version.
> ## Q12 Provenance visualization. We provide additional visualization of our learned provenances in Fig. 2(a) in the attached PDF. Note that we visualized provenances together with the training cameras in the supplementary video.
> ## Q13 “Ours” built on NeRF?
> Ours is built on SCADE [58] (c.f. Ln 232-233). The only difference from SCADE is the extra $\mathcal{L}\_{fIMLE}$ loss in Eq. (8).
> ## Q14 How sampling is done in practice.
> D is sampled every 1000 iterations following IMLE [32], and (x, t, d) are sampled every iteration.
> ## Q15 $\mathcal{L}\_{provNeRF}$ v.s. $\mathcal{L}\_{provNVS}$.
> $\mathcal{L}\_{provNeRF}$ is crucial for learning the provenance stochastic process while $\mathcal{L}\_{provNVS}$ leverages this process to enhance scene reconstruction. Ablation studies show that training without $\mathcal{L}\_{provNeRF}$ leads to slightly worse results (`Ours*` in Table S1). This is because joint optimization synergistically adapts the provenance samples to the current geometry during joint optimization. Conversely, omitting $\mathcal{L}\_{provNVS}$ results in 21.44, 0.716 and 0.349 for PSNR, SSIM and LPIPS respectively. This is inferior to the pretrained SCADE because $\mathcal{L}\_{provNeRF}$ does not improve the reconstruction using provenances.
>
> [73] Jiang, W. et. al. (2023). FisherRF, CVPR 24’

---

> > ### Comment · Reviewer_GKL5 · 2024-08-14
> > **response**
> >
> > Thanks for answering my questions in details. Most of concerns are addressed. I find this paper studing an important problem and the solution is sound and inspiring. I recommend acception.

---

### Author Rebuttal · Authors · 2024-08-07

We thank reviewers for their feedback and for finding our approach interesting (u5T3, GM1F), sound (GKL5), and novel (o98v). Our new formulation is classically motivated (GKL5), can be applied to any base NeRF models (GM1F), and leads to “a combination of traditional 3D vision-related fields to novel view synthesis” (u5T3). We also experimentally validate our provenance stochastic process in two applications that both lead to improvements over previous SotA methods (GKL5, o98v, GM1F). Below we include a summary of each response. Please see individual reviewer responses for more details.
## [GKL5, u5T3] Applying ProvNeRF to 3DGS.
We plug in ProvNeRF to a pretrained 3DGS on Scannet with additional depth supervision for convergence. We model a provenance distribution for each splat using IMLE with a shared 6-layer MLP for $H_\theta$, which takes around 30 minutes to train. After training we use it for uncertainty estimation as delineated in Sec. 5.2 (main). Figure 1 in the attached PDF shows a comparison to FisherRF [73], a recent uncertainty estimation work on 3DGS. Compared to their uncertainty map, ours shows more correlation to the depth error as highlighted by the boxed regions. Quantitatively we evaluate NLL on the three Scannet scenes shown on the right side of the same figure and show substantial improvements over FishRF. This improvement over existing literature suggests applying ProvNeRF to other representations such as 3DGS is promising. We leave further exploration of the method and applications as future works.
## [GKL5] Alternative ways to model provenances.
We experimented with other modeling strategies such as deterministic fields, gaussian mixtures, and VAEs. However, all of these methods suffer from mode collapses. Table 4 and S1 show our method’s quantitative advantage in provenance sampling and NVS co-training.
## [GKL5] How to interpret uncertainty visualization and uncertainty evaluation.
Uncertainty maps are typically measured by their correlation to depth error [19]. An ideal uncertainty map should mark regions with high-depth error with high uncertainty and vice versa. We quantitatively evaluate uncertainty maps using negative log-likelihood following prior works [51-52].
## [GKL5] Testing benchmarks.
ScanNet and T&T are standard evaluation datasets used in prior works [58] that are scene-level with varied training camera setups.
## [GKL5, GM1F, o98v] Provenance stochastic process visualization.
We visualize provenance samples on the ground truth scene surfaces in Figure 2 (a) of the attached PDF. Finally, we note that the visualized provenance directions are negative of the predicted directions for illustration. We will include this clarification and the figure in the final version.
## [GKL5] $\mathcal{L}\_{provNVS}$ v.s. $\mathcal{L}\_{provNeRF}$.
$\mathcal{L}\_{provNeRF}$ is important for learning the provenance stochastic process and $\mathcal{L}\_{provNVS}$ is crucial for improving scene reconstruction using the provenance process.
## [u5T3] Evaluation of the proposed idea.
The evaluation of ProvNeRF is split into two parts: Sec. 5.1 evaluates its NVS improvement using the additional regularizer $\mathcal{L}\_{provNVS}$. This enables the removal of common artifacts in NeRFs and improves the reconstruction metrics (GM1F, o98v). Sec. 5.2 evaluates our uncertainty estimation where we show SotA performance (o98v) compared with baselines.
## [u5T3] Results not significantly improved.
Table 1, 2, and Figure 3 in Sec. 5.1 shows $\mathcal{L}\_{provNVS}$ can remove common artifacts in indoor NeRFs (GM1F). This leads to improvements in reconstruction metrics (GM1F, o98v). We provide the rendered depth images in Figure 2 (b) of the attached PDF to demonstrate the improved geometry.
## [GM1F, o98v] Overly mathematical presentation.
Thanks for the suggestions. We aimed to present the method section with precise definitions. Instead, we will lighten the mathematical notations, move derivation details in Sec 4 to supplementary, and present the method section in an intuitive manner.
## [GM1F, o98v] Length of the provenance direction.
We use the length of the provenance direction to model its visibility (c.f. Eq.(5)). A lower direction norm usually means that the 3D location is occluded from this direction. Additionally, we made a notational mistake and represented both the normalized and unnormalized provenance directions as **d**. We will fix this in the final version.
## [o98v] Importance of provenances.
Modeling provenances in NeRF allows for analyzing NeRF’s reconstruction from traditional 3D vision (u5T3), shedding critical insights on NeRF’s convergence (u5T3, GM1F). We will better motivate provenances in the intro of the final version.
## [o98v] The method only addresses triangulation uncertainty.
While one of our applications is to model triangulation uncertainty, isolating triangulation uncertainty can benefit downstream tasks such as next-view augmentation [19]. We leave modeling other types of uncertainty (e.g., transients) as further work.
## [o98v] Stochastic Process v.s. Stochastic Field.
Thanks for the suggestion. There seems to be a terminology inconsistency for random functions with a multivariate indexing set [74-75]. But we agree that stochastic field is a better terminology as we have an $\mathbb{R}^3$ indexing set. We will change stochastic processes to fields in the final version. Note that we still use the term provenance stochastic process in rebuttal for terminology continuity.
## [GKL5, u5T3, GM1F, o98v] Typos and figures.
We thank all reviewers for their suggestions. We will fix the typos and improve the figures for better illustration and clarity in the final version.

[73] Jiang, W. et. al. (2023). FisherRF, CVPR 24’

[74] Shinozuka, M. (1987). Stochastic Fields and their Digital Simulation.

[75] Knill, O. (2009) Probability Theory and Stochastic Processes with Applications.

---

### Decision · Program_Chairs · 2024-09-25

**Decision:**

Accept (poster)

**Comment:**

The reviewers generally agree that the proposed approach is well written/motivated, has technical novelty, and is evaluated sufficiently, outperforming baselines. However, there are also shared concerns about the clarity of the theoretical presentation, the significance of the results, and difficulty interpreting visualizations. A robust rebuttal was submitted that helped to address most of the reviewer concerns. After the discussion phase, there is a shared belief that this work will be of interest to the community. As such, the AC reached a decision to accept the paper. Please take the reviewer feedback into account when preparing the camera ready version.